# New Species of *Byssosphaeria* (Melanommataceae, Pleosporales) from the Mexican Tropical Montane Cloud Forest

**DOI:** 10.3390/jof11020089

**Published:** 2025-01-24

**Authors:** Aurora Cobos-Villagrán, Abigail Pérez-Valdespino, Ricardo Valenzuela, César Ramiro Martínez-González, Isolda Luna-Vega, Lourdes Villa-Tanaca, Aída Verónica Rodríguez-Tovar, Tania Raymundo

**Affiliations:** 1Laboratorio de Micología, Departamento de Botánica, Escuela Nacional de Ciencias Biológicas, Instituto Politécnico Nacional, Prolongación de Carpio and Plan de Ayala s.n., Col. Santo Tomás, Alcaldía Miguel Hidalgo, Mexico City 11340, Mexico; cobos.fungi@gmail.com (A.C.-V.); rvalenzg@ipn.mx (R.V.); 2Laboratorio de Ingeniería Genética, Departamento de Bioquímica, Escuela Nacional de Ciencias Biológicas, Instituto Politécnico Nacional, Prolongación de Carpio and Plan de Ayala s.n., Col. Santo Tomás, Alcaldía Miguel Hidalgo, Mexico City 11340, Mexico; valdespino_abigail@hotmail.com; 3Herbario Micológico José Castillo Tovar, Instituto Tecnológico de Ciudad Victoria, Tecnológico Nacional de México, Boulevard Emilio Portes Gil No. 1301, Ciudad Victoria 87010, Tamaulipas, Mexico; cesar.ramiro.mg@gmail.com; 4Laboratorio de Biogeografía y Sistemática, Departamento de Biología Evolutiva, Facultad de Ciencias UNAM, Ciudad Universitaria, Mexico City 04510, Mexico; luna.isolda@gmail.com; 5Laboratorio de Biología Molecular de Bacterias y Levaduras, Departamento de Microbiología, Escuela Nacional de Ciencias Biológicas, Instituto Politécnico Nacional, Prolongación de Carpio and Plan de Ayala s.n., Col. Santo Tomás, Alcaldía Miguel Hidalgo, Mexico City 11340, Mexico; mvillat@ipn.mx; 6Laboratorio de Microbiología Médica y Molecular, Departamento de Microbiología, Escuela Nacional de Ciencias Biológicas, Instituto Politécnico Nacional, Prolongación de Carpio and Plan de Ayala s.n., Col. Santo Tomás, Alcaldía Miguel Hidalgo, Mexico City 11340, Mexico; avrodriguez@ipn.com

**Keywords:** Ascomycota, Dothideomycetes, taxonomy, phylogeny

## Abstract

*Byssosphaeria* Cooke is a monophyletic genus of the family Melanommataceae. The genus is characterized by ascomata smaller than 1000 µm, globose, well-developed subiculum, with a flat ostiole, and yellow-orange or reddish-brown color around the ostiole. The peridium is composed of an external layer of irregular cells followed by an internal layer of thinner cells. Clavate asci have fusiform ascospores, a hyaline-to-brown color, with one or more septa. The genus *Byssosphaeria* is composed of 29 species: saprophytes, endophytes, and parasites of woody angiosperms, and they are found in wood, leaves, and other decaying substrates. The distribution of these species is cosmopolitan, and four species have been described in Mexico. This study describes, through morphological characteristics and the phylogenetic analysis of molecular markers (ITS, SSU, LSU, *tef1-α*), four new species of *Byssosphaeria*: *B. bautistae*, *B. chrysostoma*, *B. neorhodomphala*, and *B. neoschiedermayriana*. These species are saprophytes on wood rot and are distributed in mountainous mesophilic forests from the states of Hidalgo, Puebla, and Oaxaca. The significance of this study is in the diversity of this genus in Mexico since eight species have been described.

## 1. Introduction

*Byssosphaeria* Cooke is a monophyletic genus of the family Melanommataceae [1,2,3]. It was described by Cooke and Plowright [4] and is characterized by ascomata generally smaller than 1000 µm. The ascomata are globose, sub-globose, ovoid, and turbinate, with a basal tomentose subiculum, solitary, scattered, or sometimes gregarious, and superficial with flattened ostiole. The edge of the ostiole can present colorations from yellowish-orange or reddish to even greenish [2,5]. The rest of the ascomata is black, with a coriaceous consistency, and has hyphal appendages that fuse with the developed subiculum. The peridium comprises two layers: the external layer comprises irregular cells with thick walls and an epidermoid texture, its color ranging from brown-to-dark brown, while the internal layer is formed by small, thin-walled, light-brown cells. Hamathecium is typically trabecular, thick, with long trabeculated pseudoparaphyses embedded in mucilage and anastomosing between and above the asci [6]. Asci are bitunicate, cylindrical–clavate-to-clavate, broadly rounded apically with an ocular chamber, and octospored; fusiform ascospores taper at both ends, mostly straight, sometimes slightly curved, and are hyaline or pale brown, with a single central septum, slightly constricted, smooth or slightly warty, and can be uniseriate or biseriate [2,5,7,8]. Coelomycetous asexual morphs present similarities with *Pyrenochaeta* o *Chaetophoma* [2,5,9,10,11], developing pycnidia with conidiogenous cells, referred to as phialides, that occupy the cavity, with ellipsoidal or subglobose, and hyaline conidia [5,7].

*Byssosphaeria* species are saprobes, endophytes, and parasites of woody angiosperms. These specimens are found on debarked wood, bark, fallen branches, and fallen and decaying leaves, as well as on petioles and pericarps. These species present a cosmopolitan distribution in terrestrial, freshwater, and marine habitats [2,3,5,7,12].

Currently, 29 species are recognized according to the Index Fungorum [13]. Four species have been reported in Mexico: *Byssosphaeria jamaicana* (Sivan.) M.E. Barr from the tropical montane cloud forest (TMCF) on *Quercus* fruit cupules on the ground [14] and on decomposing wood remains in an abandoned coffee plantation, both reports from the state of Veracruz [15]; *Byssosphaeria rhodomphala* (Berk.) Cooke, from the TMCF in Veracruz [15]; *Byssosphaeria schiedermayriana* (Fuckel) M. E. Barr, on decaying wood, in the *Quercus* Forest in Veracruz [15]; from the tropical cloud forest in Hidalgo [16] and Oaxaca state [17]; and finally, *Byssosphaeria xestothele* (Berk. & M.A. Curtis) M.E. Barr, on old leaves of *Loranthus crassipes* in Tamasopo, San Luis Potosí state [5], in the relict forest of *Fagus grandifolia* var. *mexicana* in Veracruz [15], and leaf litter, growing in a twig under a *Fagus grandifolia* subsp. *mexicana* tree [18]. It should be noted that *Byssosphaeria diffusa* Cooke has also been cited in Veracruz [15]; however, it is currently recognized as *Herpotrichia diffusa* (Cooke) (Ellis & Everh) [2].

The Mexican TMCF is one of the most diverse ecosystems for fungi [19,20]. This biome in Mexico has been well described by Raymundo et al. [21] and is one of the world’s most diverse areas for these taxa. Describing and understanding the fungal species in the TMCF is essential for different reasons; mainly, characterizing fungal diversity in the TMCF is relevant for forest conservation, and these forests are a source of bioactive secondary metabolites [22].

This study aimed to describe four species of *Byssosphaeria* through morphological and molecular characteristics from the Mexican TMCF.

## 2. Materials and Methods

### 2.1. Morphology

A review of the fungal collection at the ENCB-IPN herbarium was conducted. The specimens were collected from different TMCFs in Mexico, in the following physiographic provinces: Sierra Madre Oriental in the Huasteco Karst in the municipalities of Tlanchinol, Hidalgo state, and Naupan, Puebla state; in the Eje Neovolcánico, subprovince of the lakes and volcanoes of Anáhuac, in the municipality of Acaxochitlán, Hidalgo state; and the Sierra Madre del Sur, subprovince of Sierra Madre de Oaxaca, in the municipality of Santiago Camotlán, Oaxaca state. The climate is temperate and humid, and the altitudes range from 700 to 1600 m asl (meters above sea level) and from 1900 to 2100 to 4000 m asl in Santiago Camotlán. [23]. The specimens were examined using traditional mycology techniques. The ascomata were measured using a stereoscopic microscope (S9-E, Leica, Wetzlar, Germany and Stemi SV 11, Zeiss, Jena, Germany). Transverse sections were made in the midsection of the pseudothecia, which were mounted in temporary preparations with 70% alcohol and 5% KOH. These samples were observed under an optical microscope at 400× and 1000× (Zeiss K-7, Jena, Germany) to describe the following characteristics: the reaction of the peridium coloration with KOH; the thickness of the peridium; the size, shape, and diameter of the paraphyses; and the size, shape, number, and arrangement of the ascospores in the ascus. Additionally, the ascospores’ size, shape, and color were recorded based on the descriptions [5,7,15,24]. Furthermore, using the phase-contrast illumination technique, the asci and ascospores were observed at 1000× under an optical microscope (Zeiss Axiophot, Germany), adapted to the Zen 2012 capture program.

### 2.2. Extraction, Amplification, and Sequencing of DNA

Genomic DNA was extracted from herbarium specimens using the CTAB method [25]. The DNA was quantified with a Nanodrop 2000c (Thermo, Waltham, MA, USA). The nuclear genes regions, such as the internal transcribed spacer (ITS), large subunit nuclear rRNA gene (LSU), small subunit nuclear rRNA gene (SSU), and translation elongation factor 1-α (*tef1-α*), were amplified using the primers shown in Table 1. The reaction mixture for PCR was prepared in a final volume of 13 μL containing a 1x Taq DNA polymerase buffer, 0.8 mM deoxynucleotide triphosphate (0.2 mM of each), 100 ng of DNA, 20 pmol of each primer, and 2 units of GoTaq DNA polymerase (Promega, Madison, WI, USA). All the PCR reactions were performed in a Peltier Thermal Cycler PTC-200 (BIORAD, Mexico City, México). The PCR products were verified by agarose gel electrophoresis. The gels were run for 1 h at 95 V cm^−3^ in 1.5% agarose and 1x TAE buffer (Tris Acetate-EDTA). The gel was stained with GelRed (Biotium, Fremont, CA, USA), and the bands were visualized in an Infinity 3000 transilluminator (Vilber Lourmat, Eberhardzell, Germany). The amplified products were purified with the ExoSAP Purification kit (Affymetrix, Santa Clara, CA, USA), following the manufacturer’s instructions. They were quantified and prepared for the sequencing reaction using BigDye Terminator v.3.1 (Applied Biosystems, Waltham, MA, USA). These products were sequenced in both directions with the Applied Biosystems model 3730XL (Applied Biosystems, USA). The sequences of both gene strands were analyzed, edited, and assembled using BioEdit v.7.7 [26] to generate consensus sequences. These consensus sequences were compared with those deposited in GenBank at the National Center for Biotechnology Information (NCBI) using the BLASTN v.2.2.19 tool [27].

### 2.3. Phylogenetic Analysis

The sequence data retrieved from GenBank based on previous studies are listed in Table 2. The sequences were subjected to standard BLAST searches in GenBank to determine the primary identity of the fungal isolates. *Fusiconidium aquaticum* and *Fusiconidium mackenziei* [32] were used as outgroups.

The phylogenetic relationships were established by adding newly produced sequences of eight individuals of *B. bautistae*, *B. chrysostoma*, *B. neoschiedermayriana*, and *B. neorhodomphala* to reference sequences of ITS, LSU, SSU, and *tef1-α* deposited in the NCBI database (http://www.ncbi.nlm.nih.gov/genbank/, accessed on 1 October 2024). Each region was independently aligned using the online version of MAFFT v.7.490 with the L-INS-i strategy for accurate alignment [33,34]. The alignments were revised in PhyDE v.0.9971 [35], followed by minor manual adjustments to ensure character homology between taxa. The matrices were assembled for ITS with 21 taxa (689 characters), for LSU with 40 taxa (625 characters), for SSU with 24 taxa (810 characters), and for the *tef1-α* region with 26 taxa (895 characters). Six partition schemes were established: one for ITS, one for LSU, one for SSU, and three for *tef1-α*, and were created using the option to minimize stop codons with Mesquite v.3.81 [36]. The topological incongruence between partitions was examined using the incongruence length difference (ILD) test implemented in PAUP 4.0 [37] with 1000 heuristic replicates after the removal of all invariable characters. The data were analyzed using maximum parsimony (MP), maximum likelihood (ML), and Bayesian inference (BI). Maximum parsimony analyses were carried out in PAUP* 4.0a169 [38] using the heuristic search mode, with 1000 random starting replicates and TBR branch swapping, with MULTREES and Collapse on. Bootstrap values were estimated using 1000 bootstrap replicates under the heuristic search mode, each with 100 random starting replicates. Maximum likelihood analyses were conducted in RAxML v. 8.2.X [39] with the substitution model GTR + G, and bootstrap values were obtained through 1000 repetitions of nonparametric bootstrapping. Bayesian inference analysis uses the concatenate sequence to combine multiple sequences in a specified order, generating a new complete sequence. PartitionFinder v.2.1 [38,40,41] was then used to determine optimal nucleotide substitution models for each data partition. A phylogenetic tree was constructed using MrBayes v.3.2.7 [42] with the following parameter settings: the number of MCMC chains, 4 chains (1 cold chain + 3 hot chains); generations, a total of 10 million generations, with sampling every 1000 generations; initial burn-in for 25% of the total samples discarded as burn-in data; and chain convergence, with convergence assessed by an average standard deviation ≤0.01. Chain convergence was visualized in Tracer v.1.7.2 [43]. The remaining trees were used to calculate a 50% majority-rule consensus topology and posterior probabilities (PP). Trees were visualized and optimized in FigTree v.1.4.4 [44] and edited in Adobe Illustrator (Adobe Systems, Inc., San Jose, CA, USA).

**Table 2 jof-11-00089-t002:** Species names, strain numbers, isolation sources, and GenBank accession numbers for the taxa used in the phylogenetic analysis. Sequences generated in this study are shown in bold.

Fungal Species	Strain/Voucher	ITS	LSU	SSU	*tef1-α*	Reference
** *Byssosphaeria bautistae* **	**T. Raymundo 6308 ENCB Holotype**	PQ778308	PQ773902	PQ779856	PV009316	**In this study**
** *B* ** **. *bautistae***	**R. Valenzuela 16932 ENCB**	PQ778309	PQ773903	PQ779857	PV009317	**In this study**
** *B* ** **. *chrysostoma***	**T. Raymundo 6221 ENCB Holotype**	PQ778310	PQ773904	PQ779858	PV009318	**In this study**
** *B* ** **. *chrysostoma***	**A. Cobos-Villagrán 352 ENCB**	PQ778311	PQ773905	PQ779859	PV009319	**In this study**
*B. guangdongense*	ZHKUCC 22-0335	OQ449320	OQ449288	OQ449337	–	[32]
*B. guangdongense*	ZHKUCC 22-0336	OQ449321	OQ449289	OQ449338	–	
*B. jamaicana*	SMH 1403	–	GU385152	–	–	[1]
*B. jamaicana*	SMH 3464	–	GU385153	–	–	
*B. jamaicana*	SMH 3085	–	GU385154	–	–	
*B. macarangae*	MFLUCC 17 2655	MH389782	MH389778	MH389780	MH389784	[45]
*B. musae*	MFLUCC 11 0146	NR185364	NG228735	NG228698	MH581149	[46]
*B. musae*	MFLUCC 11 0182	KP744435	KP744477	–	–	
** *B* ** **. *neorhodomphala***	**E. Escudero-Leyva 190 ENCB Holotype**	PQ778314	PQ773908	PQ779860	PV009320	**In this study**
** *B* ** **. *neorhodomphala***	**T. Raymundo 4481 ENCB**	PQ778315	PQ773909	PQ779861	PV009321	**In this study**
** *B* ** **. *neoschiedermayriana***	**R. Valenzuela 16092 ENCB Holotype**	PQ778312	PQ773906	PQ779862	PV009322	**In this study**
** *B* ** **. *neoschiedermayriana***	**T. Raymundo 4199 ENCB**	PQ778313	PQ773907	PQ779863	PV009323	**In this study**
*B. poaceicola*	HFJAU10337	PP460781	PP460774	PP460766	PP475455	[47]
*B. poaceicola*	HFJAU10338	PP460782	PP460775	PP460767	PP475456	
*B. phoenicis*	ZHKUCC 21-0122	ON180685	ON180683	ON180691	ON243583	[48]
*B. phoenicis*	ZHKUCC 21-0123	ON180686	ON180684	ON180692	ON243584	
*B. rhodomphala*	SMH3086	–	GU385155	–	–	[1]
*B. rhodomphala*	SMH 4363	–	GU385156	–	–	
*B. rhodomphala*	GKM L153N	–	GU385157	–	GU327747	
*B. rhodomphala*	ANM 942	–	GU385160	–	–	
*B. rhodomphala*	SMH 3402	–	GU385170	–	–	
*B. rhodomphala*	HA 400	–	KT313008	–	–	
*B. rhodomphala*	HA 200	–	–	–	KT313006	[49]
*B. salebrosa*	SMH 2387	–	GU385162	–	GU327748	[1]
*B. schiedermayeriana*	MFLUCC 10-0100	–	KT289894	KT289896	–	[7]
*B. schiedermayeriana*	SMH 1269	–	GU385158	–	–	[1]
*B. schiedermayeriana*	SMH 1816	–	GU385159	–	–	
*B. schiedermayeriana*	SMH 3157	–	GU385163	–	GU327745	
*B. schiedermayeriana*	GKM 152N	–	GU385168	–	GU327749	
*B. siamensis*	MFLUCC 10 0099	–	KT289895	KT289897	KT962059	[7]
*B. siamensis*	MFLUCC 17 1800	MG543923	MG543914	MG543917	–	
*B. siamensis*	MFLU 18 0032	MH388334	MH376706	MH388303	MH388370	[50]
*B. siamensis*	HFJAU10336	PP460780	PP460773	PP460765	PP475454	[47]
*B. taiwanense*	MFLUCC 17 2643	MH389783	MH389779	MH389781	MH389785	[45]
*B. villosa*	GKM 204N	–	GU385151	–	GU327751	[1]
*Fusiconidium aquaticum*	KUMCC 15-0300	–	KX641894	KX641895	KX641896	[51]
*F. mackenziei*	MFLU 14-0434	–	KX611112	KX611114	KX611118	

– Absence of sequences.

## 3. Results

### 3.1. Molecular Analysis

The concatenated ITS, LSU, SSU, and *tef1-α* datasets comprise 41 taxa with 3019 characters, including gaps. The three phylogenetic analyses (MP, ML, and BI) of the datasets showed similar topologies (Figure 1). No significant conflicts (bootstrap value > 80%) were detected among the topologies obtained via individual phylogenetic analyses. The ILD test comparing the nuclear markers yielded a *p*-value of 0.12, indicating congruence among the nuclear datasets. For this reason, a combined dataset was used for the analysis. The best-fit models of the Bayesian analysis of the combined dataset were GTR + F + I + G4 for ITS, LSU, and *tef1-α*, and HKY + F + G4 for SSU. The parsimony analysis of the alignment found 1024 trees of 187 steps (CI = 0.1005, HI = 0.1004, RI = 0.2034, RC = 0.1380). We present the best RAxML tree, with a final likelihood value of 26,080.008214. The matrix had 962 distinct alignment patterns, with 3.02% undetermined characters or gaps. The estimated base frequencies were as follows: A = 0.109024, C = 0.102731, G = 0.100074, T = 0.100210; substitution rates AC = 1.000300, AG = 1.002045, AT = 1.000301, CG = 1.000844, CT = 4.000932, and GT = 1.1000000; the gamma distribution shape parameter was α = 0.000721. The standard deviation between chains in the Bayesian analysis stabilized at 0.001 after 4.5 million generations. No significant changes in tree topology trace or cumulative split frequencies of the selected nodes were observed after about 0.25 million generations and were discarded as a 25% burn-in. The phylogenetic exhibited *Byssosphaeria bautistae*, *Byssosphaeria chrysostoma*, *Byssosphaeria neoschiedermayriana*, and *Byssosphaeria neorhodomphala* and formed a monophyletic group (BS = 100%, BS = 100%, BI *p* = 1) (Figure 1).

### 3.2. Taxonomy

The new species proposed in this study are separated into four different clades, with strong bootstrap support, and affiliate with the morphological concept of *Byssosphaeria*, according to Barr [5] and Tian et al. [7]. Based on morphological, ecological, and molecular characteristics, we described these new species as follows: *Byssosphaeria bautistae*, *B. chrysostoma*, *B. neorhodomphala*, and *B. neoschiedermayriana*. 

*Byssosphaeria bautistae* Cobos Villagrán, R. Valenz., and Martínez-González & Raymundo sp. nov. (Figure 2).

Mycobank: MB856275.

Diagnosis: Differs from the other species of *Byssosphaeria* by large ascomata measuring 900–1100 × 700–1000 µm and ascospores that are olive-to-brown in color.

Type: MEXICO. Hidalgo state. Acaxochitlán municipality, in the surroundings of Acaxochitlán, town (20°09′55″ N, 98°11′47″ W), 2200 m asl, 23 September 2016, T. Raymundo 6308 (Holotype: ENCB).

Genbank: ITS PQ778308; LSU PQ773902; SSU PQ779856; *tef1-α*PV009316.

Etymology: Dedicated to Biol. Leticia Romero Bautista for her contributions to the funga of the Hidalgo State, Mexico.

Pseudothecia 900–1100 μm diameter × 700–1000 μm high, globose-to-subglobose, rounded apex, when old umbilicate, superficial, scattered, or sometimes gregarious and coria ceous; the surface is smooth at the apical part, while the rest has hyphal appendages that shape the very evident subiculum, with abundant hyphal appendages at the base up to the middle of the pseudothecia, which is hairy. The hyphae of the subiculum are black. Ostiole non-papillate-to-sparsely papillate, epapillate with a pore, surrounded by a pale yellowish-to-orange zone, in contact with 5% KOH, it releases an orange pigment that turns vinaceous. Peridium 55–90 μm thick, formed by isodiametric pseudoparenchymatous cells, reddish at the apex and dark brown-to-olive-brown elsewhere. Pseudoparaphyses 1–2 μm wide, hyaline, and trabeculated. Asci 105–125 × 11–12 μm, clavate, bitunicate, biseriate, 8-spored. Ascospores (31–) 33–33 (–36) × 5–6 μm, fusiform, olive-brown with a central septum that constricts them in the central part. In some ascospores, apices were slightly darker than the rest and surrounded by a hyaline, mucilaginous, evanescent sheath. Asexual morph: Undetermined.

Habit: gregarious on decaying wood in the TMCF.

Distribution: MEXICO. Hidalgo state.

Material examined: MÉXICO. Hidalgo state. Acaxochitlán municipality, in the surroundings of Acaxochitlán, town (20°09′55″ N, 98°11′47″ W), 2200 m asl. 23 September 2016, R. Valenzuela 16932 (ENCB), ITS PQ77830; LSU PQ773903; SSU PQ779857; *tef1-α* PV009317.

Notes: Morphologically, it is similar to *Byssosphaeria jamaicana* due to the presence of the subiculum beneath the ascomata and its gregarious growth. However, they differ because *B. bautistae* has much larger ascomata, measuring 900–1100 μm in diameter, compared to 340–550 μm [5], for Mexican specimens of *B. jamaicana*, which are similar in size at 300–500 μm [14], and 330–450 μm [15]. Another difference is in coloration: in the area around the ostiole, *B. jamaicana* is grayish-pale-to-pale brown, while *B. bautistae* is pale yellow-to-red-orange, more similar to *B. rhodomphala* or *B. schiedermayriana*. Nevertheless, these species differ in the sizes of ascomata, asci, and ascospores.

*Byssosphaeria chrysostoma* Cobos Villagrán, R. Valenz., Valdespino, Villa-Tanaca, & Raymundo sp. nov. (Figure 3).

Mycobank: MB856276.

Diagnosis: Differs from the other species of *Byssosphaeria* because it has a thick peridium of 75–150 µm, very large ascospores measuring (37–) 40–47 (–50) × 6–8 µm, and a grayish-brown color.

Type: MEXICO, Puebla state. Municipality of Naupan. On the outskirts of the town of Naupan, on the border with the municipality of Acaxochitlán (20°10′51″ N, 98°07′43″ W), 1900 m asl, 14 August 2016, T. Raymundo 6221 (Holotype: ENCB).

Genbank: ITS PQ778310; LSU PQ773904; SSU PQ779858; *tef1-α* PV009318.

Etymology: The epithet refers to the slightly golden area around the pore of the ostiole.

Pseudothecia 700–800 µm diameter × 850–950 µm high, globose-to-turbinate, with apical part completely flattened, superficial, scattered, and solitary. The surface is irregularly roughened at the apical parts, and the rest of the ascoma, the subiculum, is poorly developed. Ostiole with a flattened-to-infundibuliform apex, non-papillate, with a pore, dark brown-to-black, and with a light brown-to-slightly golden area around the pore; in reaction with 5% KOH, it changes to red and gradually loses color to a light pink-orange. Peridium 75–150 μm thick, formed by isodiametric pseudoparenchymatous cells. Asci 122–152 × 10–13 μm, clavate, bitunicate, biseriate, and hyaline. Pseudoparaphysis 1–2 μm wide, hyaline. Ascospores (37–) 40–47 (–50) × 6–8 μm, fusiform, grayish-brown, translucent, with a central septum, slightly constricted, with two guttules and sharp apices with a hyaline mucilaginous sheath. Asexual morph: undetermined.

Habit: gregarious of fallen twigs in the TMCF.

Distribution: MEXICO. Puebla state.

Material examined: MEXICO. Puebla state. Municipality of Naupan. On the outskirts of the town of Naupan, on the border with the municipality of Acaxochitlán (20°10′51″ N, 98°07′43″ W), 1900 m asl, 14 August 2016, A. Cobos-Villagrán 352 (ENCB), ITS PQ778311; LSU PQ773905; SSU PQ779859; *tef1-α*PV009319.

Notes: *B. chrysostoma* is characterized by having large ascospores. Other species with similarly sized ascospores are *B. guangdongensis*, with ascospores measuring 30–40 × 5–10 µm [32], and *B. poaceicola*, with ascospores of 32–40 × 7–8 µm [47]. *B. chrysostoma* has ascospores 10 µm longer. *B. chrysostoma* shows affinity with *B. salebrosa*, as both species possess ascomas approximately 800 µm in diameter and ascospores with similar length and width ranges. The ascospores of *B. salebrosa* are slightly broader, measuring (30–) 40–50 × (6–) 7–9 µm, compared to those of *B. chrysostoma*, which are (37–) 40–47 (–50) × 6–8 µm. However, they differ in the thickness of the peridium, which is considerably greater in *B. chrysostoma* (75–150 µm), while in *B. salebrosa*, it measures 30–35 µm, and at its widest point (at the base), 55–100 µm. Furthermore, the color of the ascoma around the ostiole is also different: in *B. chrysostoma*, it is reddish-orange-to-slightly golden around the pore, whereas in *B. salebrosa*, it is yellowish and does not lose pigmentation in KOH. Another important characteristic is the ecological aspect, as *B. salebrosa* has been reported on Acer spicatum, a tree native to North America (northeastern USA and Canada). However, it has also been found in *Vaccinium* and *Andromeda* [5]. In the case of *B. chrysostoma*, no host has been identified, as it was only found growing in the TMCF.

*Byssosphaeria neorhodomphala* Cobos Villagrán, Valdespin, R. Valenz, Rdgz. Tovar & Raymundo, sp. nov. (Figure 4).

Mycobank: MB856277.

Diagnosis: Differs from the other species by the peridium color when it is in contact with KOH.

Type: MEXICO, Oaxaca state, Villa Alta District, Municipality of Santiago Camotlán, the road to Río Blanco (17°27′59.5″ N, 96°11′04.0″ W), 750 m asl, 25 July 2013, E. Escudero-Leyva 190 (Holotype:ENCB).

Genbank: ITS PQ778314; LSU PQ773908; SSU PQ779860; *tef1-α* PV009320.

Etymology: The epithet refers to *Byssosphaeria rhodomphala* due to similar characteristics.

Pseudothecia 300–500 μm diameter × 400–600 μm high, subglobose-to-slightly turbinate, rounded at the apex, gregarious, superficial; the surface is rough-to-granular at the apical part, while the rest has hyphal appendages that form a very evident subiculum, with abundant hyphal appendages at the base up to the middle of the pseudothecia, tomentose. Ostiole non-papillate, with a central pore surrounded by a reddish-orange-to-intense red zone; the color may be localized only around the ostiole or extended to the entire middle part of the pseudothecia, while the rest of the ascoma is black. Peridium 25–65 μm thick, laterally 25–50 μm, slightly thinner at the base at 42.5 μm thick and widening towards the apex (ostiole) to 65 μm, formed by isodiametric pseudoparenchymatous cells, brown-olive towards the ostiole, reddish turning wine-to-pink color in reaction with 5% KOH. Pseudoparaphyses 1–2 μm in diameter, hyaline, branched. Asci 90–95 × 12–13 μm, fusoid, hyaline, octospores, uniseriate at the base (the first four ascospores), and biseriate towards the apex. Ascospores (18–) 20–23 (–24) × 5.5–6 (–7) μm, fusiform with obtuse apices, brown-olive-to-dark brown, with a single central septum, slightly constricted, and without a mucilaginous sheath. Asexual morph: undetermined.

Habitat: Gregarious on decaying wood in a TMCF.

Distribution: MEXICO. Oaxaca state.

Material examined: MEXICO. Oaxaca state, Villa Alta Dictrict, Municipality of Santiago Camotlán, road to Río Blanco (17°27′59.5″ N, 96°11′04.0″ W), 750 m asl,), 25 March 2013, R. Valenzuela 14871 (ENCB), T. Raymundo 4481 (ENCB), ITS PQ778315; LSU PQ773909; SSU PQ779861; *tef1-α* PV009321; 15 July 2013, T. Raymundo 4623 (ENCB), R. Valenzuela 15047 (ENCB), E. Escudero-Leyva 192 (ENCB).

Notes: Morphologically, *Byssosphaeria neorodhomphala* is similar to *B. rhodomphala* in the size of the ascomata, the thickness of the peridium, the asci, and the ascospores; although they vary by a few microns, they coincide in size ranges. However, the color around the ostiolar pore in *B. rhodomphala* has been reported as powdery, red, orange, or yellow [5], with a reddish crown [52] and an orange-reddish-to-deep red color [15]. When reviewing the type of material and specimens from Puerto Rico and Mexico, the latter authors observed that the peridium in contact with KOH released a wine-colored hue; this character has not been previously described. In *B. neorodhomphala*, the color observed is red, sometimes appearing pink. Another difference is that in *B. neorodhomphala*, the ascospores do not show the presence of a mucilaginous sheath, as described in *B. rhodomphala*. Phylogenetically, these species are not close.

*Byssosphaeria neoschiedermayriana* Cobos Villagrán, Valdespino, R. Valenz., Luna-Vega & Raymundo sp. nov. (Figure 5).

Mycobank: MB856278.

Diagnosis: Differs from the other species regarding the longer size of the ascospores, measuring 36–46 (–50) µm, in addition to the characteristic pattern around the ostiole.

Type: MEXICO, Hidalgo state, Tlanchinol Municipality, El Temazate, km. 168 of the Pachuca-Tampico highway (21°03′10″ N, 98°38′00″ W), 1500 m asl, 26 May 2012, R. Valenzuela 16092 (Holtype: ENCB).

Genbank: ITS PQ778312; LSU PQ773906; SSU PQ779862; *tef1-α*PV009322.

Etymology: The epithet refers to *Byssosphaeria schiedermayriana* due to similar characteristics.

Pseudothecia 400–770 μm in diameter, globose, rounded at the apical part, and defined by a colored area; the rest of the ascoma is black, shiny, and coriaceous, scattered-to-gregarious on the surface, with a basal subiculum. Ostiole has a flattened, non-papillate apex, reddish-orange, with lighter orange stripes arranged radially in the pore; the color in contact with 5% KOH changes to pink-fuchsia. Peridium 60–90 μm thick, with a prismatic texture, composed of two layers of cells: the outermost layer is brown, with cells measuring (12–) 18–24 × 10–17 (–19) μm and thick walls of 1–1.5 µm. The second layer comprises hyaline cells with thin walls, measuring 10–12 (–18) × 6–8 (–12) μm. Hamathecium 500–525 × 450–470 µm, with pseudoparaphyses 1–2 μm wide. Asci 142–158 (–170) × 13–14 (–15) μm, cylindrical, octosporic, and biseriate. Ascospores 36–46 (–50) × 6–7 μm, fusiform, pale brown, with a 1-transverse septum, slightly constricted septum, and a mucilaginous sheath. Asexual morph: undetermined.

Habit: gregarious on decaying wood in the TMCF.

Distribution: MEXICO. Hidalgo state.

Material examined: MEXICO. Hidalgo state, Tlanchinol municipality, Km. 168 of the Pachuca-Tampico highway (21°03′10″ N, 98°38′00″ W), 1500 m asl, 26 May 2012, R. Valenzuela 14680 (ENCB), A. Cobos-Villagrán 13 (ENCB), T. Raymundo 4199 (ENCB), ITS PQ778313; LSU PQ773907; SSU PQ779863; *tef1-α*PV009323.

Notes: *Byssosphaeria schiedermayriana* is primarily distinguished by the brightly yellow, orange, or reddish-colored area around the ostiole and ascospores with 1–5 septa surrounded by a gelatinous sheath [5,6,7]. Specimens of *B. neoschiedermayriana* have affinities with *B. schiedermayriana*; however, the size of the ascoma, asci, and ascospores is slightly larger. The ostiole’s surrounding area is also characteristically orange with lighter orange radial stripes. The color changes from an intense coral color to fuchsia upon reaction with 5% KOH. This characteristic has not been previously mentioned, as in the case of *B. rhodomphala*, where [15] a reaction with 5% KOH was observed showing a wine color in Mexican specimens and the holotype.

## 4. Discussion

*Byssosphaeria* presents high taxonomic richness in Mexico, being one of the best-studied genera of Dothideomycetes in the country. The four new species described in this study are distributed in Hidalgo, Puebla, and Oaxaca, located in the Sierra Madre Oriental, Trans-Mexican Volcanic Belt, and Sierra Madre del Sur, in mountainous areas characterized by abrupt topography and high beta diversity (Figure 6). Mexican TMCF has been considered essential for maintaining biodiversity due to the richness and endemism of fungi. Important centers of speciation and differentiation of fungi resulting from the comprehensive impacts of the upward thrust of mountains, development, and climate fluctuations in geographic history TMCFs, are the most threatened terrestrial ecosystems at the national level, so they have been classified as “habitats in danger of extinction”.

The first studies on *Byssosphaeria* were carried out only with morphological data [14,15,16,17,18], so it is necessary to review the reported specimens, as they could represent new species. Such is the case for *B. schiedermayriana*, cited by Raymundo et al. [16], which is now identified as *B. neoschiedermayriana*. Another case is that of *B. rhodomphala*, which has been considered a cosmopolitan species [5] based on morphological characters and is probably a cryptic species.

The morphological characters that provide information for the identification of *Byssosphaeria* species are the size of the pseudothecia, mainly the diameter; the length of the ascospores; and the coloration of the peridium around the ostiole in contact with KOH, as previously suggested by Chacón-Zapata and Tapia-Padilla [15]. The hosts, the environment, and the type of vegetation which the species inhabit are also important factors in their identification. In Table 3, these and other characteristics of the 33 species (including those described in the present work) can be consulted; however, very little information is available for several species.

In this study, maximum likelihood and Bayesian inference analysis on LSU, SSU, ITS, and *tef1-α* combined sequence data to elucidate the species’ phylogenetic relationships. The phylogeny of the genus *Byssosphaeria* was constructed with only 11 sequences available for the NCBI of the 29 species cited for the world (Table 2), considering those that mostly have four molecular markers (SSU, LSU, ITS, and *tef1-α*). Sequences of *Byssosphaeria villosa* (Samuels & E. Müll.) in Boise [1] were also considered. However, Li and Zhuang [52] mention that the specimen was assigned to the genus *Herpotrichia*.

The Mexican species were grouped into three different clades. On the one hand, *B. bautistae* and *B. neorhodomphala* were grouped with *B. siamensis* (described from Thailand [7,50], with support values (100 MP, 100 ML, and 1 BIPP) in separate branches. Apparently, the Asian species (China, Thailand, and Taiwan) form a single clade, along with the two species from Mexico. *B. bautistae* present larger ascomata with a diameter of 900–1100 μm and a wall thickness of 55–90 μm, while *B. siamensis* forms pseudothecia of 300–500 μm in diameter, with a wall thickness of 38–42 μm. In both species, the area around the ostiole is orange-yellowish; however, in *B. bautistae*, in contact with KOH, it releases an orange pigment that becomes vinaceous, while in *B. siamensis* this characteristic is not mentioned. As for the ascospores, *B. bautistae* has a length of (31–) 33–33 (–36) μm; in contrast, those of *B. siamensis* are 40.5–50 μm. *B. neorhodomphala* forms a basal clade with the aforementioned species, with support values of (100 MP, 100 ML, and 1 BIPP). This species forms subglobose ascomata and ascospores of (18–) 20–23 (–24) × 5.5–6 (–7) μm. Morphologically, it resembles *B. rhodomphala*; however, it is differentiated by the red-to-pinkish color of the peridium around the ostiole in contact with KOH and the absence of a mucilaginous layer on the ascospores. Phylogenetically, both species are separated, grouping into different clades.

*Byssosphaeria salebrosa*, described previously in the USA and Canada clustered with *B. chrysostoma* on separate branches, forms a clade with support values of (99 MP, 99 ML, and 1 BIPP). Both species show morphological differences, such as the color of the peridium around the ostiole. *B. chrysostoma* is light brown-to-slightly golden, which, in the presence of KOH, reacts and changes to red and gradually loses its color to pink-orange, while in *B. salebrosa,* the color of the ostiole is yellowish, and the pigment does not leach out in KOH [5].

*Byssosphaeria neoschiedermayriana* shows that it is basal to *B. jamaicana* and *B. rhodomphala*, with support values of 99 MP, 99 ML, and 1 BIPP. The three species are very similar in shape, color, consistency, and habit. Globular ascomata have a coloration around the pore of the ostiole that varies from orange to reddish, have a leathery consistency, and are gregarious. However, there are marked differences in the thickness of the peridium. It is thicker in *B. neoschiedermayriana* (60–90 μm) compared to *B. jamaicana* at 50–60 μm and *B. rhodomphala* at 20–60 μm. The asci and ascospores in *B. neoschiedermayriana* are larger: 142–158 (–170) × 13–14 (–15) μm and 36–46 (–50) × 6–7, respectively; in *B. jamaicana*, the asci are 80–120 × 12–15 μm and the ascospores are 25–35 × 7–8 μm; in *B. rhodomphala*, the asci and ascospores are smaller: (50–) 85–120 × 10–13 and (16–) 18–23 (–25) × (5–) 6–7.5 (–9) μm, respectively. In these three species, the color of the peridium around the pore of the ostiole ranges from reddish-orange-to-yellowish-orange, but in contact with KOH, they react differently: fuchsia pink in *B. neoschiedermayriana* and vinaceous in *B. rhodomphala* [15]. This character has not been described in *B. jamaicana* [5].

*Byssosphaeria* is a monophyletic genus within the family Melanommataceae, and species of this taxon are probably divergent (with a high species richness), as is the case with other members of Pleosporales genera [3,56]. All the sequences available in the NCBI databases were obtained for this study. However, the molecular data for many species have not yet been created, so the interpretation of the group’s evolutionary history is incomplete. Exploring new localities and examining morphological characters of taxonomic importance has allowed the identification of species that are new to science. Identification based solely on morphology has drawbacks, and there is the possibility of errors in taxonomic interpretations [47]. In this study, in the absence of further evidence of another type, the morphological characters used for the identification of the *Byssosphaeria* species were the size of the pseudothecia, mainly the diameter; the length of the ascospores; and the coloration of the peridium around the ostiole in contact with KOH [15]. Following Tennakoon et al. [47], the hosts, environment, and type of vegetation that the *Byssosphaeria* species inhabit were also considered relevant in the identification. These fungi have been reported in at least 15 families of vascular plants with subtropical and tropical regions distribution [2,5,7,15,24,45,47,48].

Our results support the fact that the genus *Byssosphaeria* comprises at least 33 species, of which 20 are found in America, 13 in Asia, and 2 in Europe. Most of the species are distributed on the Atlantic slope of North America. Twelve are recognized for the United States and eight for Mexico, with only *B. rhodomphala* and *B. xestothele* being shared. This group of fungi is challenging to diagnose and remains poorly studied due to the tiny size of their ascomata, typically less than a millimeter in diameter. *Byssosphaeria* species show host specificity; nearly 50% have been documented on specific hosts. Given the poor knowledge about the species of this genus, it is advisable to study them in depth.

In Mexico, such species are restricted to the trees of the TMCF, which are one of the most diverse ecosystems in the country. Mexico’s location in a biodiverse transition zone between the Neotropical and Nearctic regions is one of the leading causes of species richness in this genus.

## 5. Conclusions

This study describes four new species found in the Mexican TMCFs based on phylogenetic and morphological analyses. Characterizing fungal diversity in the TMCF is relevant for forest conservation and provides essential environmental services. Until now, the number of fungal species has been difficult to ascertain due to the imprecise nature of species identification. Three of the four new species described in this study are distributed in the Sierra Madre Oriental, a complex geologic area characterized by high beta diversity. One additional species was collected in Oaxaca in regions of abrupt topography.

It is essential to inventory and describe the fungal species of the Mexican TMCF: a threatened ecosystem classified as a habitat in danger of extinction. The loss of these montane humid forests has significantly increased in recent years. Conserving this terrestrial ecosystem is crucial to maintaining the balance and richness of this important environment. From the contribution of this work, it can be mentioned that, currently, eight species of *Byssosphaeria* have been reported in Mexico.

## Figures and Tables

**Figure 1 jof-11-00089-f001:**
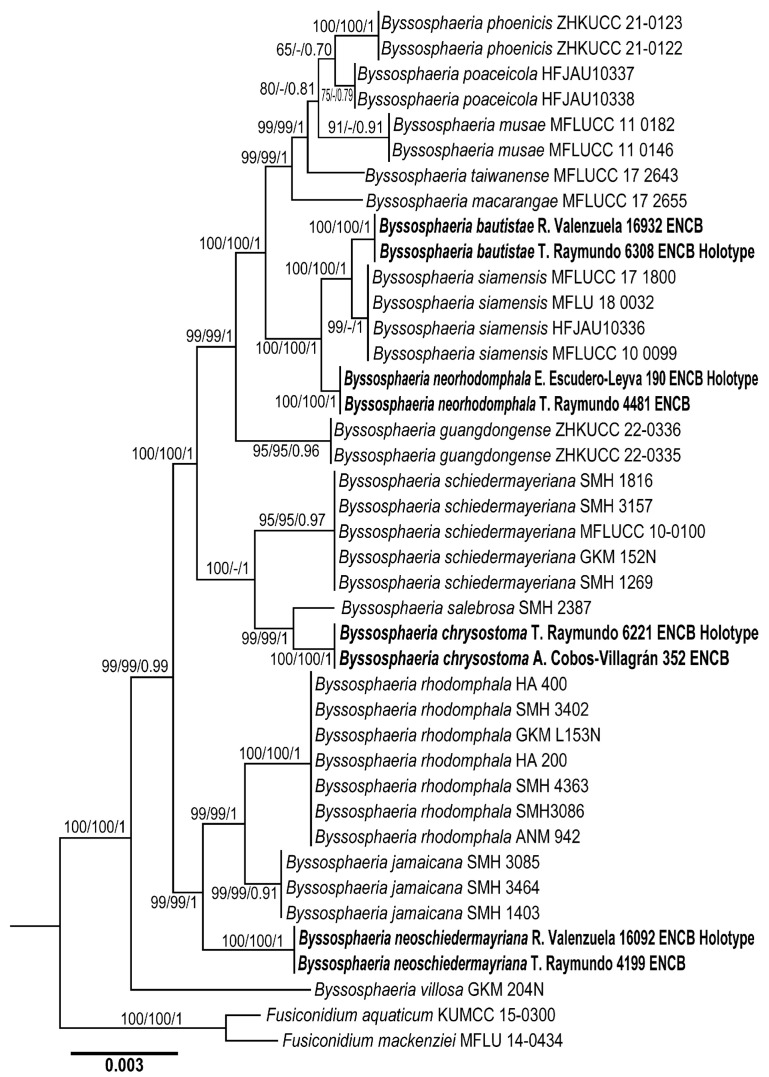
Phylogenetic reconstruction based on the concatenated ITS, LSU, SSU, and *tef1-α* sequence alignment. Maximum parsimony and Bayesian analyses recovered identical topologies concerning the relationships among the main clades of *Byssosphaeria* members. For each node, the following values are provided: maximum parsimony (MP ≥ 70%, left)/maximum likelihood bootstrap (ML ≥ 70%, middle) and the Bayesian inference posterior probability (BIPP ≥ 0.85, right). The scale bar represents the expected number of nucleotide substitutions per site.

**Figure 2 jof-11-00089-f002:**
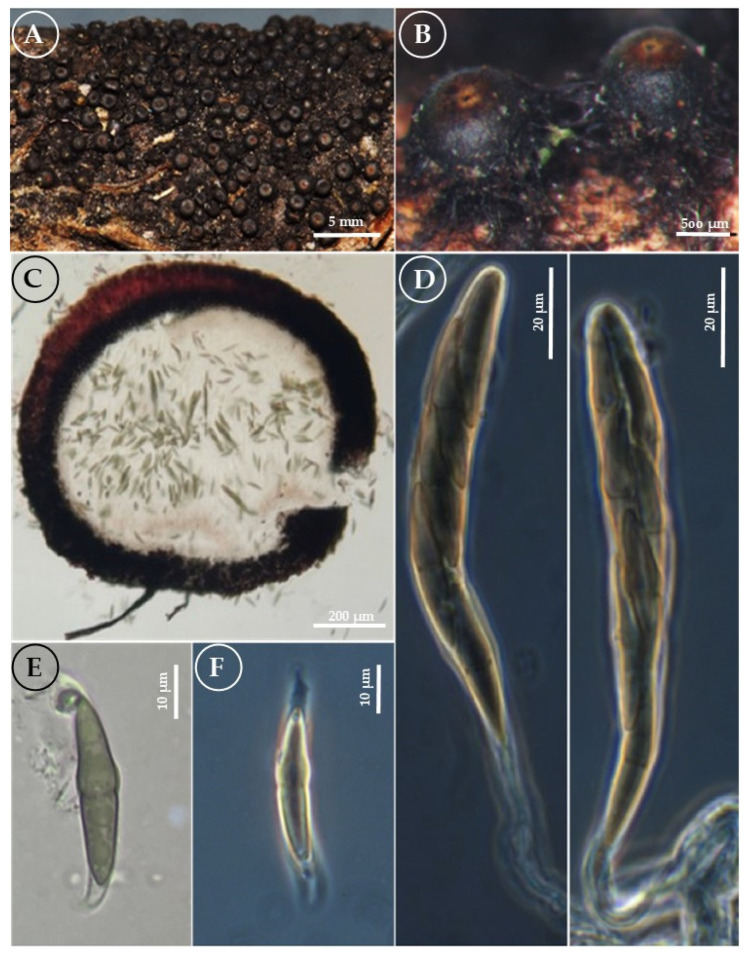
*Byssosphaeria bautistae* (T. Raymundo 6308, holotype). (**A**) Pseudothecia on the host surface/appearance of ascomata on the host; (**B**) close-up of pseudothecia and color around the ostiole; (**C**) longitudinal section of the pseudothecia and peridium; (**D**) asci; (**E**) ascospores with KOH; and (**F**) ascospores (phase-contrast).

**Figure 3 jof-11-00089-f003:**
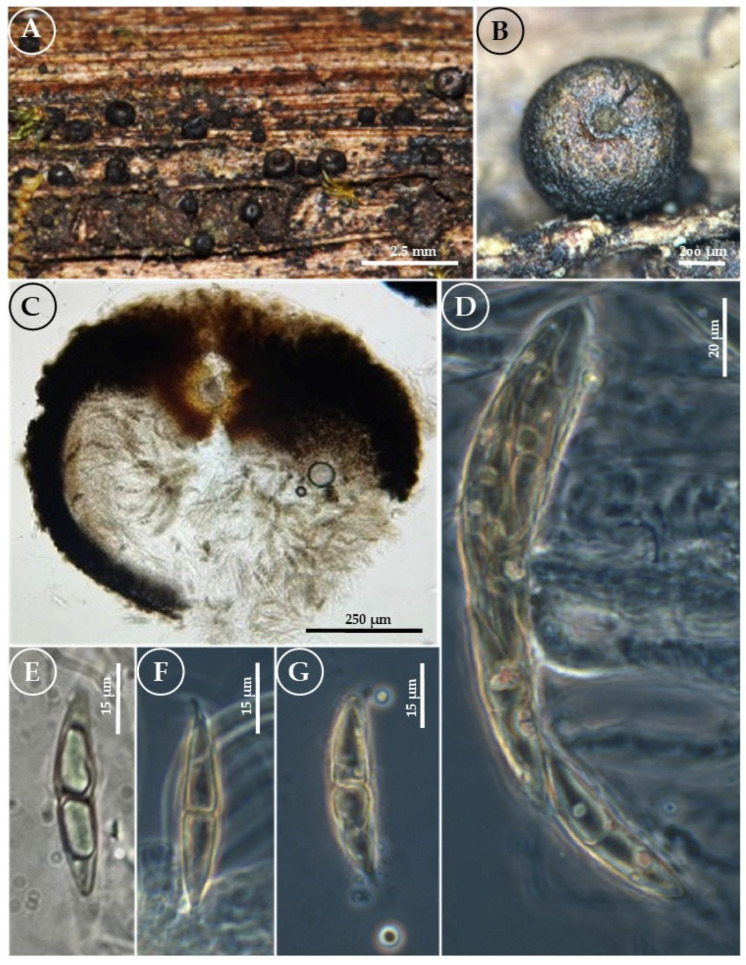
*Byssosphaeria chrysostoma* (T. Raymundo 6221, holotype). (**A**) Pseudothecia on the host surface/appearance of ascomata on the host; (**B**) close-up of pseudothecia, color around the ostiole; (**C**) longitudinal section of the pseudothecia and peridium; (**D**) asci; (**E**) ascospores with KOH; and (**F**,**G**) ascospores (phase-contrast).

**Figure 4 jof-11-00089-f004:**
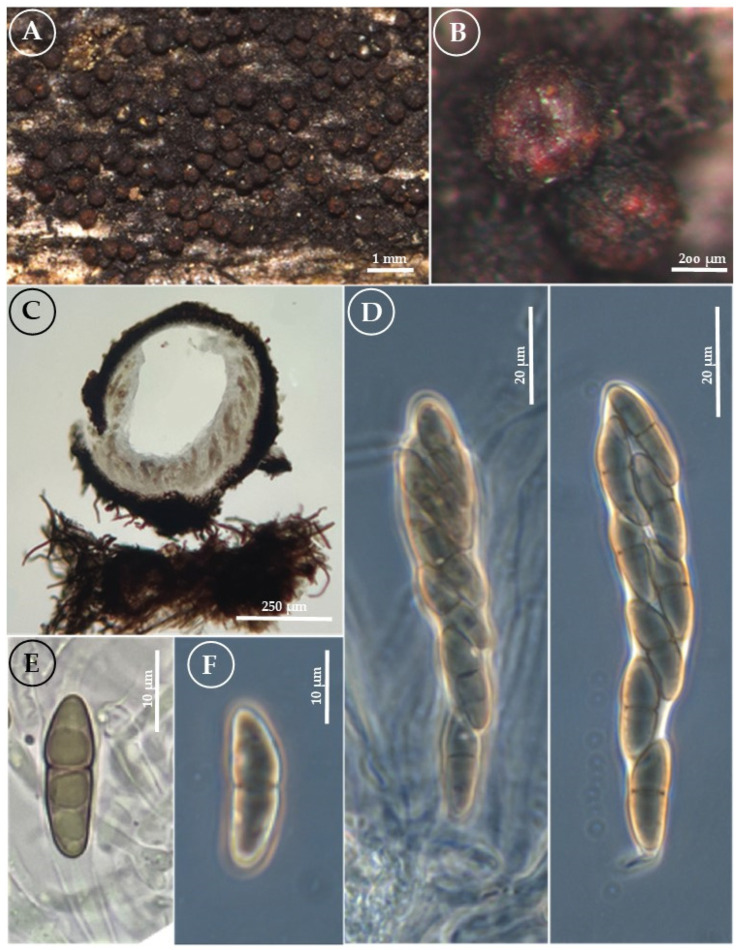
*Byssosphaeria neorhodomphala* (E. Escudero-Leyva 190, holotype). (**A**) Pseudothecia on the host surface/appearance of ascomata on the host; (**B**) close-up of pseudothecia, color around the ostiole; (**C**) longitudinal section of the pseudothecia and peridium; (**D**) asci; (**E**) ascospores with KOH; and (**F**) ascospores (phase-contrast).

**Figure 5 jof-11-00089-f005:**
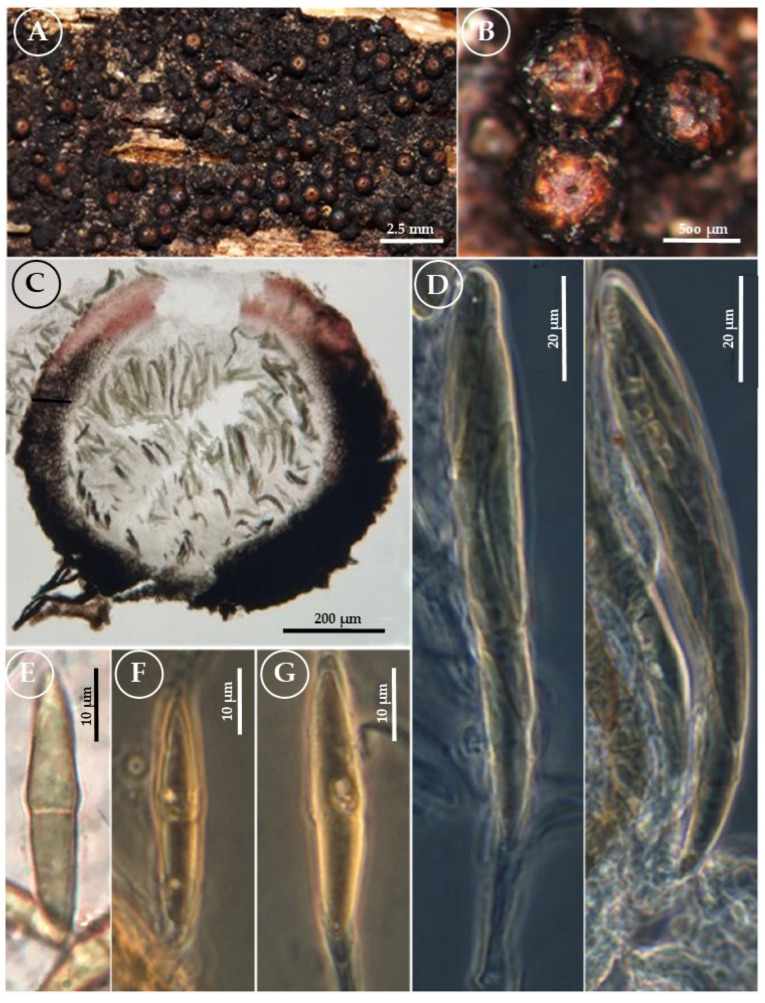
*Byssosphaeria neoschiedermayriana* (R. Valenzuela 16092, holtype). (**A**) Pseudothecia on the host surface/appearance of ascomata on the host; (**B**) close-up of pseudothecia, color around the ostiole; (**C**) longitudinal section of the pseudothecia and peridium; (**D**) asci; (**E**) ascospores with KOH; and (**F**,**G**) ascospores (phase-contrast).

**Figure 6 jof-11-00089-f006:**
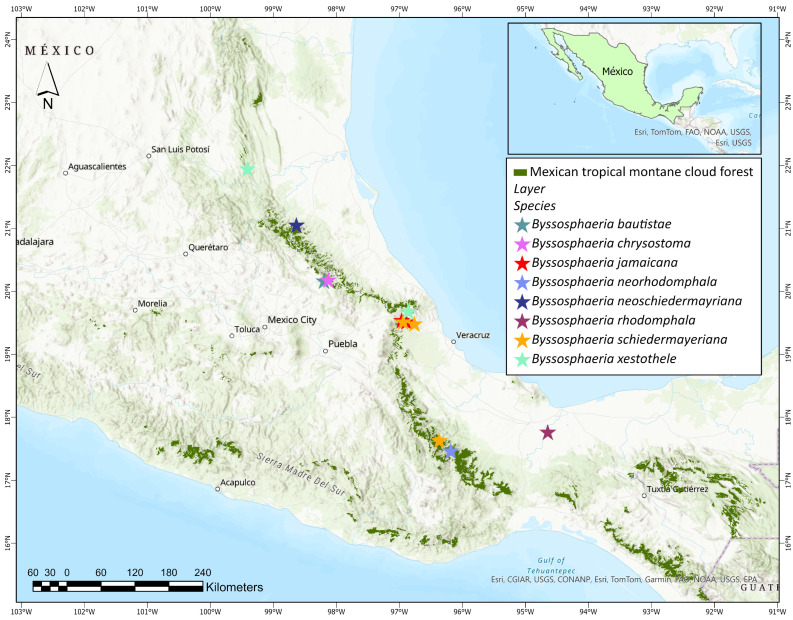
Localities of the species of *Byssosphaeria* in Mexico.

**Table 1 jof-11-00089-t001:** Primers used in this study.

Loci/Segment	Primer	Sequence 5′-3′	T(°C)	Reference
ITS	ITS5	GGAAGTAAAAGTCGTAACAAGG	58	[28]
	ITS4	TCCTCCGCTTATTGATATGC	58	
SSU	NS1	GTAGTCATATGCTTGTCTC	56	
	NS2	GGCTGCTGGCACCAGACTTGC	56	
LSU	LROR	ACCCGCTGAACTTAAGC	48	[29]
	LR3	GGTCCGTGTTTCAAGAC	48	
*tef1-α*	EF1-983F	GCYCCYGGHCAYCGTGAYTTYAT	56	[30]
	EF1-2218R	ATGACACCRACRGCRACRGTYTG	56	[31]

**Table 3 jof-11-00089-t003:** Morphological sexual characteristics of the 33 species of *Byssosphaeria*, including the 4 species studied in the present study, are highlighted in bold.

Species	Ascomata Diam × High (µm)	Peridium			Asci (µm)	Ascospores			Habitat	Type	Distribution	Reference
		Wide (µm)	Color Around the Ostiole	Color in Contact with KOH		Size Large × Wide (µm)	Number of Septa	Color				
** *B. alnea* **	220–460	40–50	Shining black, pallid	–	105–140 × 7.5–12	19.5–24 × 4–5	1–3	Light brown	On branches of *Alnus*	USA (NY)	USA, Nova Scotia, China	[5]
** *B. andurnensis* **	–	–	–	–	–	–	0	Brown	–	–	–	[13]
** *B. bautistae* **	900–1100 × 700–1000	55–90	Pale yellowish-to-orange zone	Orange pigment that turns vinaceous	105–125 × 11–12	(31–) 33–33 (–36) × 5–6	1	Olive-brown	On decaying wood	Mexico (Hgo.)	Mexico	In this study
** *B. byssiseda* **	–	–			–	–	–	–	On *Salix alba*	–	–	[13]
** *B* ** **. *chrysostoma***	700–800 × 850–950	75–150	Light brown-to-slightly golden	Red-to-light pink-orange	122–152 × 10–13	(37–)40–47(–50) × 6–8	1	Grayish-brown	On fallen twigs	Mexico (Pue.)	Mexico	In this study
** *B. epileuca* **	–	–			–	–	–	–	–	Sri Lanka	Sri Lanka	[53]
** *B. erumpens* **	420–520 × 450–550	20–30 to 70	Pallid	–	105–130 × 12–15	20–25 × 5–6	1	Dark brown	On dead stems of *Litsea* sp.	Taiwan	Taiwan, China	[24]
** *B. erythrinae* **	–	–	–	–	–	–	–	–	On dead branches of *Erythrina indica*	–	New Caledonia	[13]
** *B* ** **. *guangdongense***	480–640 × 310–500	40–70	Orange-to-light brown	–	110–160 × 10–25	30–40 × 5–10	1	Hyaline-to-brown	Saprobic on rotting branches on *Phoenix roebelenii* (Arecaceae)	China	China	[32]
** *B. hainanensis* **	182–291 × 165–280	33–58	–	–	115–130 × 7–11	10.5–13.9 × 3.5–5.2	1	Light clear brown	On decaying wood	China	China	[52]
** *B. holophaea* **	500	–	–	–	–	22–24 × 8	2	–	On branches	USA (PA)	USA	[53]
** *B. imposita* **	–	–	–	–	–	25 × 6	0	Brown	On dead branches	USA (PA)	USA	
** *B. jamaicana* **	340–500	50–60	Dark reddish brown	–	80–120 × 12–15	25–35 × 7–8	1–3	Light-to-clear brown	On decorticated rooting wood	Jamaica	Jamaica, Puerto Rico, Trinidad & Tobago, China, Mexico	[5]
** *B. juniperi* **	–	–	–	–	100–120 × 20	30–35 × 10–12	–	–	On bark of *Juniperus monosperma*	USA (CO)	USA	[54]
** *B. macarangae* **	60–120 × 80–150	20–30	Black	–	100 × 7–10	20–25 × 4–5	1 with 2 euseptate	Hyaline	On decaying wood of *Macaranga tanarius*	Taiwan	Taiwan	[45]
** *B* ** **. *musae***	450–630 × 430–540	35–80	Orange-to-yellow	–	(120–) 125–135 (–145) × (11.5–) 12–14 (–17)	30–33 (–36) × (4–) 5–6	1 (–3)	Hyaline-to-light brown	On leaf sheath of *Musa* sp. (Musaceae)	Thailand	Thailand	[46]
** *B* ** **. *neorhodomphala***	300–500 × 400–600	25–65	Reddish-orange-to-intense red	Reddish to a wine color to pink	90–95 × 12–13	(18–) 20–23 (–24) × 5.5–6 (–7)	1	Brown-olive-to-dark brown	On decaying wood	Mexico (Oax.)	Mexico	In this study
** *B* ** **. *neoschiedermayriana***	400–770	60–90	Reddish orange	Pink-fuchsia	142–158 (–170) × 13–14 (–15)	36–46 (–50) × 6–7	1	Pale brown	On decaying wood	Mexico (Hgo.)	Mexico	In this study
** *B. oviformis* **	1000–1500	100–130	Dull black	–	120–130 × 7–9	25–30 × (2.5–) 3–3.5	1	Hyalinae	On blackened decorticated wood	Jamaica	Jamaica, Hong Kong, China	[5]
** *B. pardalios* **	–	–	–	–	–	8.8 long	0	Brown	–	–	–	[13]
** *B. phoenicis* **	580–625 × 600–650	90–110	Reddish orange	–	100–160 × 10–15	25–30 × 5–7	1 (–3)	Pale brown-to-pale olivaceous	On dead petioles of *Phoenix roebelenii* (Arecaceae)	China	China	[48]
** *B* ** **. *poaceicola***	550–650 × 600–800	30–45	Orange-to-yellow	–	165–180 × 12–15	32–40 × 7–8	1	Hyaline-to-pale brown	On dead stem of *Arundo pliniana* (Poaceae)	China	China	[47]
** *B. picta* **	–	–	–	–	–	20–22 × 8–10	0	Brown	On decorticated wood	India	India	[53]
** *B. purpureofusca* **	–	–	–	–	–	–	0	Brown	On branches of *Quercus*	USA (PA)	USA	
** *B. rhodomelaena* **	–	–	–	–	–	10 × 6	0	Brown	On rotten wood.	USA (Carolina & PA)	USA	
** *B* ** **. *rhodomphala***	220–500	20–60	Red, orange, or yellow pulverulence	Pigment leaching	(50–) 85–120 × 10–13	(16–) 18–23 (–25) × (5–) 6–7.5 (–9)	1(–3)	Light brown	On wood and periderm of various trees	USA (OH)	Cosmopolitan	[55]
** *B. rubiginosa* **	–	–	–	–	–	20–24 × 4–6	2	Hyaline-to-pale brown	–	USA (NY)	USA	[53]
** *B* ** **. *salebrosa***	440–800	30–35 to 55–100 (in base)	Yellowish	Non-leaching pigment	120–150 × 13–16.5	(30–) 40–50 × (6–) 7– 9	1–3 (–5)	Hyaline-to-light brown	On woody substrates, on dead roots of *Acer spicatum* (Sapindaceae), and on branches of *Vaccinium* or *Andromeda*	USA (NY)	USA, Canada	[5]
** *B* ** **. *schiedermayeriana***	500–825	50–100	Red or orange	–	(80–) 100–150 × 12–15	(25–) 32–42 × 5–8 (–9)	1–3 (–5)	Light brown	On varied substrates, rotting logs and branches, endocarps of coconut, culms, and petioles on wood or cord in greenhouses. On rotten branches of *Sambucus nigra*	Austria	Cosmopolitan	
** *B. semen* **	400–600 × 330–550	40–84	Pallid	–	80–110 × 9–12	20–30 × 3.5–4.5 (–6)	1 (–3)	Hyaline-to-light brown	On decaying petioles of *Sorbus* sp., rotting hardwood	USA (NY)	USA	
** *B* ** **. *siamensis***	501–692 × 561–720	39–42	Orange-to-yellow	–	112–148 × 10–16	40.5–50 × 7–11	1 (–3)	Hyaline-to-pale yellow	On decaying wood and unidentified host	Thailand	Thailand	[7]
** *B* ** **. *taiwanense***	450–500 × 460–540	35–50 to 75–85 (in base) to 75–95 (wide near ostiole)	Orange-to-yellow	–	(120–) 125–150 (–155) × (11.6–) 12–14 (–14.8)	30–35 × 7–8	1	Hyaline-to-light brown	On decaying wood of *Macaranga tanarius*	Taiwan	Taiwan	[45]
** *B. xestothele* **	330–440 × 330–550	30–52	Reddish	–	70–100 × 9–12	20–26 × 4.5–6	1–3	–	On fallen branches of *Cornus florida* or old leathery leaves	USA (SC)	USA, Mexico	[5]

## Data Availability

The original contributions presented in the study are included in the article, further inquiries can be directed to the corresponding author.

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
