# Peer review of "New Species of Byssosphaeria (Melanommataceae, Pleosporales) from the Mexican Tropical Montane Cloud Forest"

_jof, 2025, doi:10.3390/jof11020089_

Round 1
Reviewer 1 Report
In their manuscript entitled “New species of Byssosphaeria (Melanommataceae, Pleospora) from the Mexican tropical montane cloud forest”, Cobos-Villagrán and co-authors describe four new species of Byssosphaeria from the Mexican tropical montane cloud forest. The manuscript is well organized and written. There are some issues which the authors should address:
1) The manuscript should provide access to the nucleotide sequence data used for phylogenetic analysis, allowing other researchers to verify and reanalyze the data.
2) Figures are crucial for understanding the research results. I recommend checking the clarity of figure 2, 3, 5 and 6.
3) I suggest adding a taxonomic identification key to clarify the morphological relationships among the 33 species of Byssosphaeria.
4) The discussion section in the article is insufficient, it is suggested to add more content.
Minor issues:
1. “tef1-É‘” need to be italicized throughout the entire text.
2. The SSU annealing temperatures in Table 1 are 63 and 53 ℃, please unify them.
3. The holotype of different species in Figure 1 should be highlighted.
4. Which specimen should the morphological data belong to in the captions of Figures 2, 3, 4, and 5? It should be indicated.
5. Line 493-494, Delete “Please add:”.
6. Line 628, "Hu1, D.-M." should be "Hu, D.-M."
Author Response
Dear Reviewer,
Thank you for taking the time to review our manuscript. We appreciate your insightful comments and have addressed them as detailed below. All changes in the manuscript are highlighted in green.
Major Comments:
- GenBank Accession Numbers:
- Comment: The manuscript should provide access to the nucleotide sequence data used for phylogenetic analysis, allowing other researchers to verify and reanalyze the data.
- Response: We have updated the manuscript to include the available GenBank accession numbers for ITS, LSU, and SSU sequences. The tef1-α sequence is currently being processed and will be added once available.
- Image Quality:
- Comment: Figures are crucial for understanding the research results. I recommend checking the clarity of figure 2, 3, 5 and 6.
- Response: The images have been embedded in the manuscript in TIFF format to enhance quality. Additionally, they have been provided as separate files for your review.
- Taxonomic Key:
- Comment: I suggest adding a taxonomic identification key to clarify the morphological relationships among the 33 species of Byssosphaeria
- Response: It is not possible to create a taxonomic key because many Byssosphaeria species lack complete information; as seen in Table 2, several morphological characters are not mentioned in the original descriptions.
- We acknowledge the limitation in creating a comprehensive taxonomic key due to incomplete information on Byssosphaeria species. This constraint is discussed in the manuscript, particularly in relation to Table 2, where several morphological characters are absent from original descriptions.
- Discussion Expansion:
- Comment: The discussion section in the article is insufficient, it is suggested to add more content.
- Response: The discussion section has been enhanced with additional insights in lines 398–400, 414–416, and 438–441. Furthermore, lines 476–503 have been revisited and enriched to provide a more comprehensive analysis.
Detailed Comments:
- Italicization of tef1-α:
- We have italicized tef1-α in the following sections to adhere to scientific conventions:
- Lines 33, 111, 138, 144, 146, 176, 207, 415, 418
- Table 1
- Table 2
- Table 1:
- Standardized the primer annealing temperature to 53°C.
- Corrected primer sequences for accuracy.
- Holotype Indication in Figures 2–5:
- We have indicated the holotypes in Figures 2–5 to provide clarity:
- Line 263
- Line 322
- Line 423
- Line 427
- Line 527:
- Removed the placeholder text "Please add."
- Line 664:
- Deleted the extraneous numeral "1" following "Hu."
Line-by-Line Revisions:
- Line 24: Corrected the correspondence email address by removing the dash.
- Line 27: Replaced the term "pore" with "ostiole" for accuracy.
- Line 42: Substituted "The genus Byssosphaeria" with "It" to avoid repetition.
- Line 43: Corrected "or" to "to" for proper context.
- Lines 140–141: Added the phrase "with the L-INS-i strategy for accurate alignment" to specify the alignment method.
- Line 144: Removed the word "while" after "(810 characters)" for clarity.
- Line 144: Deleted "which were" after "tef1-α" to improve sentence structure.
- Line 177: Added abbreviations in parentheses "(MP, ML, and BI)" for Maximum Parsimony, Maximum Likelihood, and Bayesian Inference methods.
- Lines 179–183: Specified the models used for Bayesian analysis for each molecular marker.
- Line 226: Enhanced the description by adding "when old umbilicate."
- Line 396: Replaced "humid oak forest" with "TMCF" (Tropical Montane Cloud Forest) for precision.
- Table 3: Removed double spaces between "B." and the specific epithets: B. erumpens, B. jamaicana, B. picta.
- Line 462: Deleted "The phylogeny of" before "Byssosphaeria neoschiedermayriana shows" to avoid redundancy.
- Line 542: Changed the semicolon to a comma after the surname "Huhndorf."
- Lines 547–550, 552, 554, 565, 566, 650, 655–657, 659–662, 664, 673, 683: Replaced hyphens with periods to correct punctuation.
- Line 548: Removed the double space between "Y." and "M."
- Line 554: Added a comma after "Yacharoen."
- Line 559: Included the DOI: https://doi.org/10.1017/s0953756299008679 for reference.
- Line 560: Added a space before "Wang."
- Line 569: Provided the total number of pages, as multiple pages were consulted.
- Line 576: Removed the parentheses around the year "2024."
- Lines 581–582: Corrected the title of the manuscript to: Catálogo de los ascomicetos del bosque mesófilo de montaña de Tlanchinol, Hidalgo (México).
- Line 599: Changed the en dash to an em dash for proper punctuation.
- Line 606: Corrected the issue number to "13, 1."
- Line 610: Changed the en dash to an em dash for proper punctuation.
- Line 614: Removed the double space before "Abdel-Wahab."
- Line 655: Deleted the double space after "D.Q." and before "Doilom."
- Line 656: Removed the double space before "Jayawardena."
- Line 657: Deleted the double space after "I.C."
- Line 658: Added a space after "Tibpromma" and "Udayanga."
- Line 658: Removed the double space before "Wijayawardene."
- Lines 659, 660, 662: Deleted the double spaces before "Zeng," "Hofstetter," "Kang," and "Zhao."
- Line 668: Corrected "Phillips, A.J.l.;" to "Phillips, A.J.L.;"
- Line 680: Added a space after "Li."
- Line 689: Added a new reference cited in the discussion section.
We believe these revisions enhance the clarity and accuracy of our manuscript. Thank you again for your valuable feedback.

Reviewer 2 Report
A manuscript is presented describing four new species of the genus Byssosphaeria from tropical montane cloud forests (TMCF) in Mexico, using morphological and molecular analyses as part of what is known as integrative taxonomy. Integrative taxonomy plays a crucial role in advancing our understanding of biodiversity. This comprehensive approach is particularly significant for conservation efforts, as it enables the identification of distinct species, especially in the context of biodiversity from poorly explored regions. Therefore, this represents a valuable contribution to mycology. The included figures and tables, as well as the distribution maps and phylogenetic trees, help contextualize the findings. The manuscript has high potential for publication.
However, there are areas that could be improved to strengthen the scientific impact and clarity of the study.
Although robust phylogenetic analyses are included, the discussion on the position of the new species and their relationship to previously described species is somewhat superficial. Expanding the discussion to include factors or processes that explain the divergence of Mexican species compared to those from other regions would be beneficial. Nonetheless, only 11 of the recognized species currently have molecular data in GenBank. This limits conclusions about evolutionary relationships and the overall robustness of the study. While the aim of this work is not to produce a global monograph of the genus, this limitation could still be addressed in the discussion.
The Materials and Methods section is fairly comprehensive but lacks specific details on the use of bioinformatics tools, such as the criteria for selecting evolutionary models or the alignment parameters used. Including this information would be advisable.
The discussion would also benefit from more information on the ecology and biogeography of the species, to allow for comparisons with others. The limited information provided on this aspect is practically confined to the last paragraph of the Results section.
In Table 2, it is necessary to include the GenBank accession numbers for the samples. Currently, they appear as "in process," but this cannot be published in this way.
The bibliography is generally well-prepared, though some inconsistencies in formatting are evident. For instance, in reference 22, the page range is separated by a short dash ("–") when it should be a long dash ("–"). Please review this.
Author Response
Dear Reviewer,
Thank you for your valuable feedback on our manuscript. We have addressed your comments and made the necessary revisions, which are highlighted in green within the manuscript. Below is a summary of the changes made:
Major Comments:
- Methods Section:
- Comment: The Materials and Methods section is fairly comprehensive but lacks specific details on the use of bioinformatics tools, such as the criteria for selecting evolutionary models or the alignment parameters used. Including this information would be advisable.
- Response: We have revised the methods section to incorporate the requested details. Specifically, changes have been made in lines 147–149 and 155–164 to provide clearer methodology.
- We have provided a comprehensive description of the tools used and the evolutionary models applied in our study.
- Lines 140–141
- Lines 147–149
- Lines 155–164
- In lines 179–182, we have specified the models used for the Bayesian analysis of each molecular marker.
- Expansion of the Discussion:
- Comment: The discussion would also benefit from more information on the ecology and biogeography of the species, to allow for comparisons with others. The limited information provided on this aspect is practically confined to the last paragraph of the Results section.
- Response: The discussion section has been enhanced with additional insights in lines 398–400, 414–416, and 438–441. Furthermore, lines 476–503 have been revisited and enriched to provide a more comprehensive analysis.
Detailed Comments:
- GenBank Accession Numbers for ITS, LSU, and SSU Sequences:
- We have updated Table 2 to include the GenBank accession numbers for the ITS, LSU, and SSU sequences.
- The tef1-α sequence is still being processed and will be added upon completion.
- Correction of Reference Inaccuracies:
- We have corrected inaccuracies in the references to ensure accuracy and consistency.
Line 542: Changed the semicolon to a comma after the surname "Huhndorf."
Lines 547–550, 552, 554, 565, 566, 650, 655–657, 659–662, 664, 673, 683: Replaced hyphens with periods to correct punctuation.
Line 548: Removed the double space between "Y." and "M."
Line 554: Added a comma after "Yacharoen."
Line 559: Included the DOI: https://doi.org/10.1017/s0953756299008679 for reference.
Line 560: Added a space before "Wang."
Line 569: Provided the total number of pages, as multiple pages were consulted.
Line 576: Removed the parentheses around the year "2024."
Lines 581–582: Corrected the title of the manuscript to: Catálogo de los ascomicetos del bosque mesófilo de montaña de Tlanchinol, Hidalgo (México).
Line 599: Changed the en dash to an em dash for proper punctuation.
Line 606: Corrected the issue number to "13, 1."
Line 610: Changed the en dash to an em dash for proper punctuation.
Line 614: Removed the double space before "Abdel-Wahab."
Line 655: Deleted the double space after "D.Q." and before "Doilom."
Line 656: Removed the double space before "Jayawardena."
Line 657: Deleted the double space after "I.C."
Line 658: Added a space after "Tibpromma" and "Udayanga."
Line 658: Removed the double space before "Wijayawardene."
Lines 659, 660, 662: Deleted the double spaces before "Zeng," "Hofstetter," "Kang," and "Zhao."
Line 664: Deleted the extraneous numeral "1" following "Hu."
Line 668: Corrected "Phillips, A.J.l.;" to "Phillips, A.J.L.;"
Line 680: Added a space after "Li."
Line 689: Added a new reference cited in the discussion section.
Line-by-Line Revisions:
- Line 24: Corrected the correspondence email address by removing the dash.
- Line 27: Replaced the term "pore" with "ostiole" for accuracy.
- Lines 33, 111, table 1, 138, 144,146, table 2, 176, 207, 415, 418: We added in italics tef1-α
- Line 42: Substituted "The genus Byssosphaeria" with "It" to avoid repetition.
- Line 43: Corrected "or" to "to" for proper context.
- Table 1: Standardized the primer annealing temperature to 53°C and corrected primer sequences.
- Lines 140–141: Added the phrase "with the L-INS-i strategy for accurate alignment" to specify the alignment method.
- Line 144: Removed the word "while" after "(810 characters)" for clarity.
- Line 144: Deleted "which were" after "tef1-α" to improve sentence structure.
- Line 177: Added abbreviations in parentheses "(MP, ML, and BI)" for Maximum Parsimony, Maximum Likelihood, and Bayesian Inference methods.
- Lines 179–183: Specified the models used for Bayesian analysis for each molecular marker.
- Line 226: Enhanced the description by adding "when old umbilicate."
- Line 263, 322, 423, 427: We have indicated the holotypes in Figures 2–5 to provide clarity:
- Line 396: Replaced "humid oak forest" with "TMCF" (Tropical Montane Cloud Forest) for precision.
- Table 3: Removed double spaces between "B." and the specific epithets: B. erumpens, B. jamaicana, B. picta.
- Line 462: Deleted "The phylogeny of" before "Byssosphaeria neoschiedermayriana shows" to avoid redundancy.
- Line 527: Removed the placeholder text "Please add."
We believe these revisions enhance the clarity and accuracy of our manuscript. Thank you again for your valuable feedback.
